# Analysis of Efficacy-To-Safety Ratio of Angiogenesis-Inhibitors Based Therapies in Ovarian Cancer: A Systematic Review and Meta-Analysis

**DOI:** 10.3390/diagnostics13061040

**Published:** 2023-03-09

**Authors:** Laurentiu Simion, Vlad Rotaru, Ciprian Cirimbei, Daniela-Cristina Stefan, Mirela Gherghe, Sinziana Ionescu, Bogdan Cosmin Tanase, Dan Cristian Luca, Laurentia Nicoleta Gales, Elena Chitoran

**Affiliations:** 1Department of Surgery, “Carol Davila” University of Medicine and Pharmacy, 050474 Bucharest, Romania; 2General Surgery and Surgical Oncology Department I, Bucharest Institute of Oncology “Prof. Dr. Al. Trestioreanu”, 022328 Bucharest, Romania; 3Nuclear Medicine Department, Bucharest Institute of Oncology “Prof. Dr. Al. Trestioreanu”, 022328 Bucharest, Romania; 4Thoracic Surgery Department, Bucharest Institute of Oncology “Prof. Dr. Al. Trestioreanu”, 022328 Bucharest, Romania; 5Medical Oncology Department, Bucharest Institute of Oncology “Prof. Dr. Al. Trestioreanu”, 022328 Bucharest, Romania

**Keywords:** angiogenesis inhibitors, VEGF inhibitors, ovarian cancer, progression-free survival, toxicity, overall survival, meta-analysis, systematic review, randomized controlled trials, FDA (Food and Drugs Administration) approval of cancer drugs

## Abstract

(1) Background: Among new anti-angiogenesis agents being developed and ever-changing guidelines indications, the question of the benefits/safety ratio remains unclear. (2) Methods: We performed a systematic review combined with a meta-analysis of 23 randomized controlled trials (12,081 patients), evaluating overall survival (OS), progression free survival (PFS) and toxicity (grade ≥ 3 toxic effects, type, and number of all adverse effects. (3) Results: The analysis showed improvement of pooled-PFS (HR, 0.71; 95% CI, 0.64–0.78; I^2^ = 77%; *p* < 0.00001) in first-line (HR, 0.85; 95% CI, 0.78–0.93; *p* = 0.0003) or recurrent cancer (HR, 0.62; 95% CI, 0.56–0.70; *p* < 0.00001) and regardless of the type of anti-angiogenesis drug used (Vascular endothelial growth factor (VEGF) inhibitors, VEGF-receptors (VEGF-R) inhibitors or angiopoietin inhibitors). Improved OS was also observed (HR, 0.95; 95% CI, 0.90–0.99; *p* = 0.03). OS benefits were only observed in recurrent neoplasms, both platinum-sensitive and platinum-resistant neoplasms. Grade ≥ 3 adverse effects were increased across all trials. Anti-angiogenetic therapy increased the risk of hypertension, infection, thromboembolic/hemorrhagic events, and gastro-intestinal perforations but not the risk of wound-related issues, anemia or posterior leukoencephalopathy syndrome. (4) Conclusions: Although angiogenesis inhibitors improve PFS, there are little-to-no OS benefits. Given the high risk of severe adverse reactions, a careful selection of patients is required for obtaining the best results possible.

## 1. Introduction

Significant improvements are continuously made in the treatment of ovarian carcinoma, but this disease continues to place a tremendous burden on healthcare systems, especially in countries with a low index of human development. GLOBOCAN 2020 statistics indicate that ovarian cancer remains third in incidence and second as cause of death between the genital neoplasms. In Central and Eastern Europe, ovarian cancer has the highest incidence rate in the world (10.7/100,000) and a mortality rate of 5.6/100,000 [1]. At present, the guidelines promote optimal cytoreductive surgery and polychemotherapy consisting of platinum agents and taxanes as the standard of care. Although most patients will experience remission under therapy, 80% will relapse within 18 months [2]. The relapse is usually diagnosed in advanced stages, despite the encouraging response to platinum-based first-line therapy, and the recurrence usually shows increased chemo-resistance, leading in most cases to death [3,4]. Thus, new agents, such as angiogenesis inhibitors and poly-ADP-ribose polymerase (PARP) inhibitors, were introduced as treatment options. Regardless of the initial factors that lead to the neoplastic transformation of cells, due to the increased metabolic need and secondary tissular hypoxia, cancers require neo-angiogenesis for their progressive growth and metastasis, with cancer cells being able to stimulate surrounding normal cells to secret molecules with signaling properties in angiogenesis pathways [5,6]. Angiogenesis in solid tumors is a well-known fact, and targeting the pathways that regulate angiogenesis is suggested as a potential therapeutic approach [5,7]. After the initial paper by Folkman in 1971 [5] that shifted the therapeutic paradigm from targeting the tumor cell to an anti-angiogenetic approach, a new field of study in oncology emerged. Over the years, numerous discoveries have been made identifying several angiogenetic factors, understanding the regulation of pathological angiogenesis process in tumors and ultimately developing new drugs that block pathological tumoral vessel formation. After the establishment of VEGF as the principal mediator in tumoral angiogenetic pathways [8], targeting VEGF or its receptors VEGF-R has become the pivot in research for the development of antiangiogenetic agents. Clinical trials (number ranging in the thousand) demonstrated the benefits of anti-angiogenesis drugs. Numerous molecules have been identified, developed and approved for cancer therapy [9,10,11,12], as well as for secondary use as treatment for ocular neo-vascular diseases which share angiogenetic pathways and signaling molecules with carcinomas [11,13,14,15,16,17,18]. These therapies are credited with improved progression-free survival (PFS) and improved overall survival (OS), but those last results are inconstant. Due to the high costs, as well as the frequency and severity of therapy-specific adverse effects, in low- and middle-income countries, there is a tendency toward the careful consideration of the efficacy–safety ratio. In Romania, these therapies are fully refunded through national healthcare programs, and as a result, at the Oncology Institute of Bucharest, around 50 patients with ovarian cancers are treated each year with anti-angiogenesis drugs. Although we observed improved survival, some patients experience severe adverse reactions and extreme alterations of QoL, leading us to the conclusion that more in-depth analyses of security profiles are warranted. In Romania, we see no reduction in the incidence for ovarian cancer [19]; thus, the therapeutic option for these cases continues to be of concern.

The scope of our study is to provide an extensive review of the efficacy–safety ratio of angiogenesis-inhibitors-based therapies in ovarian cancer, thus providing a better evaluation and selection tool for clinicians.

## 2. Materials and Methods

### 2.1. Search Strategy

A systematic search of the existing literature, according to PRISMA guidelines [20], was performed in multiple international databases: PubMed, Embase, Web of Science, Cochrane Library and ASCO and ESMO abstract database, from establishment up to 1 May 2022. The first criterion used was that the publications contained a reference to ovarian cancer; thus, the following syntax was used: (ovar OR ovarian OR ovary) AND (neoplasm OR cancer OR carcinoma OR malignant OR tumor). The second criterion used was that the publications contained a reference to angiogenesis or antiangiogenetic therapy—(vascular endothelial growth factor OR angiogenesis inhibitor OR VEGF OR VEGFR OR VEGF-R OR anti-VEGF OR VEGF-target OR anti-angiogenic OR anti-angiogenesis OR antiangiogenetic). The third criterion was that the publications contained a reference to a specific anti-angiogenetic drug; thus, we used the following syntax: (bevacizumab OR avastin OR cediranib OR AZD2171 OR recentin OR aflibercept OR VEGF trap OR AVE0005 OR zaltrap OR sorafenib OR nexavar OR pazopanib OR Votrient OR trebananib OR AMG386 OR nintedanib OR BIBF 1120 OR tyrosine kinases inhibitors OR TIE OR AXL OR FLT OR sunitinib OR SU11248). We used a limitation in searching for articles written in English.

### 2.2. Study Selection

References were prepared by using Mendeley Reference Manager [21]. After removing duplicate records, an extensive title and abstract screening process was performed, and records with irrelevant focus were removed.

We included adult patients with histopathological confirmation of ovarian cancer regardless of histology and regardless of treatment settings (first-line, recurrence, or maintenance), receiving anti-angiogenetic drugs. The experimental arm needed to be compared to standard chemotherapy or placebo, with PFS and/or OS being reported by hazard ratio with 95% CI. PFS, OS and adverse effects were our outcomes of interest, but we did not limit the inclusion if a study did not report all three. The study design of trials included were Phase II/III randomized controlled. Non-randomized controlled studies or Phase I trials were also excluded due to the fact that they were not considered high-quality statistical data. Book chapters, reviews and case reports were also excluded for the same reason. Studies using cyclophosphamide as a control arm were excluded due to the fact that the superiority of carboplatin/paclitaxel over cyclophosphamide regimens was long-time established by prior trials [22].

The PICOS structure (Population-Intervention-Comparator-Outcomes-Study design) shown in Table 1 was used for including studies in our meta-analysis [23].

The remaining articles were obtained as full text and were reviewed independently by two authors, and discrepancies were the subject of discussion between all authors. The review was not registered and has no external funding.

### 2.3. Data Extraction

From each trial, we selected data pertaining to name of primary author, year of publication, study phase, patient selection criteria, drug regimens on experimental and controlled arms of study, sample size, different bias factors and outcomes intended for use in meta-analysis (PFS, OS and adverse reactions—type of event, total number of events, number, and type of grade ≥ 3 adverse reactions).

### 2.4. Assessment of the Bias Risk

The bias risk was evaluated on six domains used by the Cochrane Collaboration’s tool [24]: bias of selection, performance, detection, attrition, reporting and other types of bias. Details regarding the risk of bias are presented in Figure 1a,b. Risk was categorized as being high, unclear, or low and was color coded.

### 2.5. Statistical Analysis

Pooled hazard ratios (HRs) for progression free and overall survival were calculated using a generic inverse variance in RevMan 5.4.1 software [25]. Risk ratios (RR) for adverse reactions were also calculated using 95% CI. I^2^ type statistic was used in order to see the statistical heterogeneity. When a high heterogeneity among the studies was observed, a random model was used for statistical analysis. For I^2^ < 40%, indicating low probability of heterogeneity, we used a fixed model. Random models were used when high heterogeneity was observed. A sensitivity analysis was performed by visually assessing the gross asymmetry of funnel plots, verifying lack of publication bias. Heterogeneity among studies was investigated further by producing Galbraith plots for each analysis with NCSS 2023 software, thus verifying the consistency of results.

## 3. Results

### 3.1. Literature Search

After the review of the literature, we identified *n* = 9754 records concerning the utilization of antiangiogenetic therapies for ovarian cancer. Another 54 additional records were found after reference review. Duplicate records (*n* = 2183) were removed, and after title and abstract screening, 7514 records were excluded: irrelevant focus (*n* = 5127), non-randomized controlled/Phase I studies (*n* = 443), reviews or case reports (*n* = 1908) and other reasons (*n* = 39). The remaining 119 records were obtained as full-text articles and assessed. Another 97 studies were excluded due to lacking outcomes of interest or being previous versions or exploratory outcomes of studies already included. In the end, 23 randomized controlled trials (RCTs) were included. The literature search flowchart is presented in Figure 2.

### 3.2. Studies Characteristics

A total of 12,082 patients were included (adults with confirmed ovarian cancer, having therapies including anti-angiogenetic agents and compared to other drug regimens without such agents or placebo). The 23 RCTs have a publication date between 2011 and 2022, and they evaluated seven inhibitors of angiogenesis (bevacizumab, six; pazopanib, five; trebananib, four; nintedanib, two; sorafenib, two; cediranib, two; and aflibercept, one). In the 2019 Tewari study, there were two experimental arms, which were both included in our study, the difference being in the maintenance therapy (one arm using bevacizumab—BEVm; and one arm using placebo—PLm). A similar situation was encountered in the Ledermann 2016 study, but we did not separately use the experimental arms due to the lack of HRs for OS and PFS between the control and PLm arms of the study. The general characteristics, the references and the summary of outcomes from the included trial are included in Table 2 and Table 3.

### 3.3. Analysis of Survival: Overall and Progression-Free

In our study the anti-angiogenetic regimens were credited with an important improvement of PFS over the control arm (HR, 0.71; 95% CI, 0.64–0.78; I^2^ = 77%; *p* < 0.00001). Improved OS was also observed, but the results had a lower *p*-value (HR, 0.95; 95% CI, 0.91–0.99; I^2^ = 0; *p* = 0.02).


**
*Treatment settings analysis:*
**


**PFS:** Five studies reported data about PFS in first-line therapy, 14 in recurrent disease and 3 in maintenance. Benefits of survival until progression were observed in all subjects in first line (HR, 0.85; 95% CI, 0.78–0.93; *p* = 0.0003) and recurrent cancer (HR, 0.62; 95% CI, 0.56–0.70; *p* < 0.00001) settings. No significant improvement of PFS was observed in maintenance settings (HR, 0.89; 95% CI, 0.65–1.22; *p* = 0.47). For the recurrent cancers, the PFS improvement did not vary according to platinum-sensitivity (platinum-sensitive recurrence—HR = 0.58, 95% = CI 0.49–0.69, and *p* < 0.00001; platinum-resistant recurrence—HR = 0.56, 95% CI = 0.42–0.75, and *p* < 0.00001, respectively)—Figure 3. Six studies were used for PFS analysis in P-S R disease, and four for P-R R.

**OS:** When analyzing the OS in different treatment setting, the results were more heterogenic—significant improvements were observed in recurrent cancers (both in P-S R and P-R R—HR = 0.88, 95%CI = 0.79–0.99, and *p* = 0,03; HR = 0.78, 95% CI = 0.65–0.94, and *p* = 0.01, respectively). However, when employed as first-line or maintenance therapy, no improvement was observed (HR, 0.95; 95% CI, 0.66–1.37; *p* = 0.78)—Figure 4.


**
*Analysis by type of drugs used:*
**


Different action mechanisms of the antiangiogenetic drugs may have different effects. We tried to establish that by evaluating PFS and OS for three distinct action mechanisms—vascular endothelial growth factor inhibitors (VEGF inhibitors such as bevacizumab and aflibercept), inhibitors of VEGF receptors (VEGF-R inhibitors such as pazopanib, cediranib, nintedanib and sorafenib) and angiopoietin inhibitors such as trebananib.

**PFS:** All three groups experienced improvements on PFS: VEGF inhibitors—HR = 0.66, 95% CI = 0.54–0.80, and *p* < 0.00001; VEGF-R inhibitors—HR = 0.72, 95% CI 0.63–0.82, and *p* < 0.0001; and angiopoietin inhibitors—HR = 0.82, 95% CI = 0.69–0.99, and *p* = 0.04 (Figure 5).

**OS:** No significant differences of overall survival were observed between experimental and control arms (Figure 6).


**
*Heterogeneity and publication bias:*
**


We tried to evaluate if the heterogeneity of studies influenced the PFS/OS results and performed a sensitivity analysis by visual evaluation of asymmetry of the funnel plot for the 23 included trials and found no gross asymmetry, indicating absence of publication bias (Appendix A). In addition, radial (Galbraith) plots were created for each analysis as a supplement to forest plots, thus demonstrating the consistency of the results (Appendix A).

### 3.4. Adverse Events

We analyzed the adverse effects reported for the included studies. For the Karlan and Tewari studies, which included two experimental arms, the safety analyses were performed by comparing the two experimental arms together with the control arm. Our analysis included 31 adverse events that are considered to be associated with anti-angiogenetic therapy for which RR were calculated (Appendix A Appendix A). Hypertension was found to have a higher risk for the patients on angiogenesis inhibitors (RR, 4.26; 95% CI, 1.61–11.26; *p* = 0,0003; I^2^ = 99%). Similar results were found for hemorrhagic events (RR, 1.94; 95% CI, 1.00–3.76; *p* = 0.05; I^2^ = 77%) and for thromboembolic events (RR, 1.47; 95% CI, 1.12–1.92; *p* = 0.006; I^2^ = 2%). However, only arterial thromboembolism was statistically linked to the anti-angiogenetic therapy (RR, 2.38; 95% CI, 1.52–3.70; *p* = 0.0001; I^2^ = 4%). The same is not true for venous thromboembolism (RR, 1.13; 95% CI, 0.74–1.72; *p* = 0.58; I^2^ = 51%). A significantly increased risk was also seen for proteinuria (RR, 4.52; 95% CI, 1.93–10.55; *p* = 0.0005; I^2^ = 88%), gastro-intestinal perforations (RR, 2.86; 95% CI, 1.36–6.04; *p* = 0.006; I^2^ = 0%), infections (RR, 1.28; 95% CI, 1.07–1.53; *p* = 0.008; I^2^ = 0%), ascites (RR, 2.06; 95% CI, 1.65–2.57; *p* < 0.00001; I^2^ = 35%), anemia (RR, 0.86; 95% CI, 0.75–1.99; *p* = 0.03; I^2^ = 86%), neutropenia (RR, 1.20; 95% CI, 1.03–1.40; *p* = 0.02; I^2^ = 89%), leucopenia (RR, 1.23; 95% CI, 1.05–1.45; *p* = 0.01; I^2^ = 62%), thrombocytopenia (RR, 1.81; 95% CI, 1.43–2.29; *p* < 0.00001; I^2^ = 82%), nausea (RR, 1.28; 95% CI, 1.10–1.49; *p* = 0.002; I^2^ = 81%), vomiting (RR, 1.24; 95% CI, 1.04–1.47; *p* = 0.01; I^2^ = 51%), diarrhea (RR, 1.92; 95% CI, 1.46–2.53; *p* < 0.00001; I^2^ = 90%), dyspnea (RR, 1.25; 95% CI, 1.08–1.45; *p* = 0.003; I = 0%), hypokalemia (RR, 1.91; 95% CI, 1.45–2.51; *p* < 0.00001; I^2^ = 0%), pain (RR 1.12; 95% CI1.03–1.21; *p* = 0.0009; I^2^ = 0%) and headache (RR, 1.73; 95% CI, 1.34–2.22; *p* < 0.00001; I^2^ = 61%). No significant correlation was seen for the following symptoms and the anti-angiogenetic therapy: reversible posterior leukoencephalopathy syndrome (RR, 1.18; 95% CI, 0.18–7.94; *p* = 0.86; I^2^ = 0%), pyrexia (RR, 0.96; 95% CI, 0.75–1.23; *p* = 0.76; I^2^ = 2%), wound related issues (RR, 1.57; 95% CI, 0.96–2.56; *p* = 0.07; I^2^ = 24%), fatigue (RR, 1.12; 95% CI, 0.99–1.27; *p* = 0.06; I^2^ = 81%), hypomagnesemia (RR, 1.69; 95% CI, 0.72–3.93; *p* = 0.23; I^2^ = 46%), anorexia (RR, 1.53; 95% CI, 0.84–2.78; *p* = 0.16; I^2^ = 85%), constipation (RR, 1.08; 95% CI, 0.96–1.21; *p* = 0.19; I^2^ = 19%), alopecia (RR, 1.07; 95% CI, 0.97–1.19; *p* = 0.18; I^2^ = 0%), rash (RR, 1.34; 95% CI, 0.79–2.27; *p* = 0.287; I = 85%), abdominal pain (RR, 1.05; 95% CI, 0.95–1.17; *p* = 0.34; I^2^ = 0%) and back pain (RR, 0.92; 95% CI, 0.62–1.37; *p* = 0.70; I^2^ = 65%). Grade ≥ 3 adverse reaction risk was increased in therapeutic arms of our study (RR, 1.37; 95% CI, 1.17–1.60; *p* < 0.00001; I^2^ = 97%)—Figure 7.

## 4. Discussion

### 4.1. Interpretation of Results

Our study demonstrated that anti-angiogenetic drugs not only can improve progression-free survival in ovarian cancer in a significant way but also can increase the risk of common adverse reactions of all grades. The benefits regarding OS are more uncertain, being observed only in specific types of patients and tumors. We observed contradictory results for OS—the pooled OS for all trials showed a significant improvement of survival favoring the angiogenesis inhibitors, but we could not find a significant improvement when studying different treatment setting or types of anti-angiogenetic drugs separately. None of the therapeutic categories (VEGF blockade, VEGF-R inhibitors and angiopoietin inhibitors) were associated with improved OS. As first-line therapy, we could prove no significant correlation with improved OS. The possible reasons for this are as follows: (1) Most cases of ovarian cancer that benefit from first line chemotherapy are advanced (among the six trials included studying first-line therapy only one enrolled stage I/II cases [43]). (2) Three of the twenty-three RCTs [26,33,35] allowed patients in the control arm to receive anti-angiogenetic salvage therapy after disease progression, thus diminishing the OS differential between experimental and control arms. (3) High-risk ovarian cancer is more likely to have a maximal response when treated with anti-angiogenesis agents as first-line.

The ICON7 trial [43] considered as high-risk cancers FIGO IV, inoperable or sub-optimally resected (>1 cm residual disease) FIGO III disease. The AGO-OVAR 12 [31], TRINOVA3 [52] and GOG-0218 [48] trials all shared the same definition for high-risk tumors. However, the OS values between these trials were unconcordant: ICON7 showed that bevacizumab in high-risk tumors is associated with benefits of overall survival (HR, 0.78; 95% CI, 0.63–0.97; *p* = 0.01) [43]. On the contrary, AGO-OVAR 12 showed that an improved OS is obtained in standard chemotherapy rather than in the nintedanib-treated group (HR, 1.14; 95% CI, 0.89–1.45) [31]. The GOG-0218 study showed that, in advanced high-risk cancers, the concurrent use of bevacizumab with chemotherapy, followed by a bevacizumab maintenance arm, is optimal (HR, 0.72; 95% CI, 0.53–0.97) [48]. The 2022 network meta-analysis by Helali et al. also demonstrated that the use of bevacizumab concurrently with chemotherapy, followed by maintenance with bevacizumab until disease progression in chemo-naive disease, is associated with the highest probability of improvements of PFS and PFS [53]. Moreover, the same meta-analysis ranked the effects of different anti-angiogenetic drugs in high-risk chemo-naive disease (bevacizumab concurrent + maintenance > nintedanib concurrent + maintenance > trebananib concurrent + maintenance > standard-of-care carboplatin/paclitaxel). All mentioned trials showed no OS benefit favoring the usage of anti-angiogenetic drugs for chemo-naive non-high-risk cancers. The OS in chemotherapy-naive non-high-risk cancers can vary, thus making it difficult to propose any clinical recommendations for the usage of angiogenesis inhibitors in this disease setting, but most studies infer a probable lack of efficacy of angiogenesis inhibitors when used in the chemo-naive non-high-risk ovarian carcinoma [53].

In recurrent treatment settings, our study proved the benefits of anti-angiogenesis drugs regardless of the platinum sensitivity of disease. For both platinum-sensitive and platinum-resistant cancers, the PFS and OS were improved, so angiogenesis inhibitors can represent a valid treatment option. Helali et al. [53] ranked the angiogenesis inhibitors by probability of benefit in recurrent epithelial ovarian cancer. For the platinum-resistant group, they found that the concurrent chemotherapy and pazopanib has the best chance of improving OS, followed by the combination of chemotherapy with sorafenib. For the platinum-sensitive group, they demonstrated the lack of OS benefits when adding anti-angiogenetic drugs to standard chemotherapy, but the association of bevacizumab or cediranib to chemotherapy in the maintenance stage can improve PFS. Helali et al. also suggested that PARP inhibitors, in addition to chemotherapy, are the best option for platinum-sensitive recurrent disease [53]. Other studies suggested that PARP inhibitors in combination with antiangiogenetic agents may be a better therapeutic option than monotherapy with antiangiogenetic agents in platinum-sensitive cancers [54].

When analyzing the OS benefit of antiangiogenetic drugs used as maintenance therapy, we found no improvements.

The OS benefits are difficult to evaluate since most trials included did not report the OS based on PFI (platinum free interval), and the low sample size in some studies (MITO 11 [45] and TRIAS [27]) may influence results.

The results suggest that PFS is not a viable surrogate evaluation for OS response. The FDA approval process of various cancer drugs for solid tumors use surrogate endpoints (PFS) rather than clinical outcomes (OS). The definition of the FDA for clinical outcomes is “a direct measure of benefit of an intervention in a trial”. A predictive substitute for clinical outcomes is called a surrogate endpoint. PFS is frequently used as a surrogate endpoint for OS in solid tumors [55,56], and the usage is growing every day. Many cancer drugs received FDA approval based on surrogate endpoints such as PFS, rather than mature OS data. Among the cancer drugs that have received regulatory approval, 57% proved not to have an OS benefit after the data were available [57]. Given the results of our study, we consider that the usage of a surrogate endpoint for clinical outcomes (PFS) in the process of drug approval may need to be reconsidered, as PFS proved not to be a reliable substitute for OS. PFS is a suboptimal surrogate for OS, and this fact was highlighted by several different articles and studies [53,58,59,60].

The absence of reliable predictive biomarkers for response to angiogenesis inhibitor-based therapy makes it difficult to conduct a proper selection of patient categories that benefit most from anti-angiogenetic drugs. A series of markers have been suggested as response predictors: highAng1/lowTie2 values were associated with PFS benefits in experimental bevacizumab arm of ICON 7 patients [61]. The ICON 7 trial also validated the gene signature proposed by Gourley et al. to define the angiogenetic molecular subtypes of ovarian tumors [62]. In the serum samples from ICON 7 trial, Collinson et al. [63] identified several biomarkers that are predictive of therapeutic response: fms-like tyrosine kinase-4, mesothelin and acid-α1 glycoprotein. This resulted in identifying a patient subset likely to benefit from antiangiogenetic drugs. Patients exhibiting a positive signature profile had a median PFS improved by 5.5 months (significant *p*-value of 0.001). GOG-218 was the source of several articles identifying several potential predictors for response to anti-angiogenetic therapy: plasmatic concentrations of VEGF and VEGF-R2 [64], tumor VEGF-A expression [65], micro-vessel density (MVD) [66] and IL-6 levels [67]. Although the identification of these predictive biomarkers is encouraging, they need to be validated by more extensive trials in order to be incorporated into guideline recommendations for practicians. Oxidative-stress-related genes and factors have also been demonstrated to play a significant role in ovarian tumors as a predictive factor of response to angiogenesis inhibitor therapy in ovarian neoplasms [68,69,70]. It is known that OS affects tumor chemoresistance by specific mutations of important redox enzymes [71]. Oxidative stress factors have been identified in circulation among patients with malignant ovarian tumors [72]. However, the role of oxidative-stress-related prediction of response to therapy factors still needs to be investigated by trials. Other authors have suggested that non-coding RNAs play a role in angiogenesis in gynecological cancers [73].

Since for advanced ovarian cancer the treatment is mostly palliative in nature, and given the myriad of potentially severe adverse effects and dubitable OS benefits, the question of quality of life (QoL) of patients treated with anti-angiogenetic agents should be an important indicator of therapeutic success. At present, there are very few studies focusing on QoL in patients treated with angiogenesis inhibitors [74,75]. This aspect needs to be evaluated further by clinical trials. Our study demonstrated that some adverse effects that can significantly affect the QoL are more frequent when angiogenesis inhibitors are used. The anti-angiogenetic agents are associated with a higher incidence of gastro-intestinal perforations, infections, thromboembolic and hemorrhagic events.

When debating the efficacity–toxicity ratio of angiogenesis inhibitors, one more concern is the cross results when employed in conjunction with other another class of targeted therapies such as inhibitors of poly-ADP-ribose polymerase (PARP), especially since an ever-increasing number of trials have been published testing this therapeutic combination. This combination of drugs has a reported synergic effect, since angiogenesis inhibitors lead to hypoxia, which has been shown to induce homologous recombination repair deficiency by a downregulation of homologous recombination-repair genes. This effect results in an increased sensitivity to PARP inhibitors of tumoral cells [76,77,78]. Liu et al. [40] showed a significant improvement of PFS when treating the platinum sensitive ovarian recurrent cancer with cediranib plus olaparib, when compared to olaparib monotherapy. The combination also increases the OS in patients negative for a germline mutation of BRCA1/2. There are also data that suggested an improvement of PFS when adding olaparib maintenance in patients with advanced EOC who have received standard first-line chemotherapy plus bevacizumab—Phase II study conducted by Ray-Coquard in 2019 [54]. This combination of angiogenesis and PARP inhibitors can represent a future direction for research. Another small Phase I study (Lorusso et al., from 2020) focused on the association between bevacizumab and a different PARP inhibitor (rucaparib) and found no safety concerns about the combination [79]. The MITO 25 Phase II randomized study will investigate further rucaparib as maintenance in combination or without bevacizumab for patients with newly diagnosed FIGO III-IV ovarian cancer [80]. Mirza et al. [81] published a Phase I trial on combination niraparib and bevacizumab in platinum sensitive epithelial ovarian cancer which showed no safety concerns.

Anti-angiogenetic therapies, although constantly improving PFS, show suboptimal clinical effect [82,83]. This may be explained by the triggering of survival selection of hypoxic cells in the center of tumor [6]. Moreover, disruption of a specific angiogenic pathway may provoke compensatory reactions through compensatory production of alternative factors with roles in angiogenesis [84,85,86,87,88,89,90], leading to resistance to single-target therapeutic approaches. We therefore see an unmet need for the development and evaluation of novel strategies in order to mitigate the shortcomings of current therapeutic strategies by employing concurrent therapies that target multiple angiogenetic pathways.

### 4.2. Comparison with Other Studies

Our study consisted of 23 RCTs, thus making it the largest and a more recent meta-analysis combined with an extensive systematic review performed about anti-angiogenetic drugs and their usage in ovarian cancer, including 12,082 patients. Over time, several systematic reviews and meta-analyses demonstrating outcome variabilities were carried out on angiogenesis-inhibitor randomized controlled trials [53,91,92,93,94,95,96,97,98,99,100,101]. One of the latest (Guo) [92] included 22 RCTs published between 2011 and 2019, with 11,254 patients meeting the inclusion criteria. Our study included one more study (Duska et al., 2022). Although Helali et al. [53] included 23 RCTs, they focused on epithelial ovarian cancer, omitting studies such as Liu and Lederman’s (2016) [39,40], that were included in our study, in which we also included serous or endometrioid histology. They included the Matulonis 2019 (KEYNOTE 100) study [102] on the effects of pembrolizumab (a PD-1 inhibitor) in ovarian cancer, which we consider outside of the scope of this paper. The Hall et al., 2020 [103] study was not included due to the fact that cyclophosphamide was used as a treatment line rather than carboplatin/paclitaxel—the superiority of combination carboplatin–paclitaxel over the cisplatin–cyclophosphamide combination was demonstrated by a previous trial [22], and the usage of cyclophosphamide may falsely alter the PFS results in favor of anti-angiogenetic drugs. The most recent meta-analysis on the subject of inhibitors of angiogenesis in ovarian-tumor patients we came across is the one performed by Zhang et al. [104], referring only to recurrent ovarian tumors and containing 13 RTCs, which were all included in our study. The results of the Zhang et al. meta-analysis had similar results to ours, indicating both an OS and a PFS benefit of anti-angiogenetic drugs in recurrent neoplasm.

Angiogenesis inhibitors are associated with an increased PFS over all patient groups and all categories of drugs used, results similar to previous studies. In our study, the OS was only improved by the studied therapy for recurrent ovarian cancer, not influencing cases in first line or maintenance settings. The improved OS observed in other studies when using VEGF inhibitors or angiopoietin inhibitors was not observed in our study.

Most of previous meta-analyses carried out on angiogenesis inhibitors for patients with ovarian cancer focused on survival parameters, with toxicity being ignored. For a better understanding of the ratio of efficacy to safety of angiogenesis inhibitors, we also analyzed the therapy-specific adverse effects. By calculating the RR for 31 therapy-specific cases, we observed an increased risk for hypertension, hemorrhagic and thromboembolic events, proteinuria, gastro-intestinal perforations, infections, ascites, anemia, neutropenia, leucopenia, thrombocytopenia, nausea, vomiting, diarrhea, dyspnea, hypokalemia, headache and pain. For all grade-three-or-higher adverse reactions, there was an increased risk in the experimental arms of trials. There were no significant increased risks found in the following adverse events: venous thromboembolism, pyrexia, fatigue, hypomagnesemia, anorexia, constipation, alopecia, rash, abdominal pain and back pain. Two of the adverse effects specific to the angiogenesis inhibitors (wound-related issues and reversible posterior leukoencephalopathy syndrome) were proved not to be significantly linked to the therapy.

### 4.3. Strengths and Study Limitations

The strength of our study is the broad inclusion of 12,081 patients treated with seven different angiogenesis inhibitors, as this allowed us to perform a comprehensive analysis of survival parameters (PFS and OS) and therapy-specific toxicity. To the best of our knowledge, this is the largest meta-analysis on anti-angiogenetic drugs in ovarian cancer to this date. The RCTs included enrolled all stages of ovarian cancer and different treatment settings, thus allowing us to evaluate the role of the anti-angiogenetic therapy as first-line, as maintenance or as secondary therapy for recurrent disease. Furthermore, the subgroup analyses of different mechanism anti-angiogenetic drugs may allow us to make a prediction of the type of anti-angiogenetic therapy more likely to obtain the maximum benefit. As a side effect of the broad inclusion criteria, regardless of tumor type, stage of disease, drug regimens, number of previous lines of therapy, response to previous therapy, duration of follow-up and so on, the heterogeneity was increased, giving raise to the possibility of different results when focusing on more narrow patient types or more specific drug regimens (our study included VEGF blockade, VEGF-R inhibitors and angiopoietin inhibitors).

The high heterogeneity across studies is the most important limitation of our analysis. This high heterogeneity among trials may be explained by the inconsistency of characteristics of patients enrolled and also of disease particularities (stage of disease, pre- and postoperative tumoral burden, number of therapeutical lines used prior to inclusion in trial and histology of tumors). Different definitions for endpoints, different methodology and different sample size between studies and the fact that different anti-angiogenetic drugs were used may also account for heterogeneity. Another limitation of our study is the fact that our analysis is based on published data, rather than on actual patient records, thus making it prone to bias. Moreover, some of the studies included had an increased bias risk because of issues with randomization/blinding. The lack of registration of a Trial Sequential Analysis (TSA) protocol may also be considered a limitation of our study since some conclusions of our study may not have a sufficient effect size in order to be unlikely affected by further trials.

## 5. Conclusions

Our study showed that anti-angiogenetic agents can improve the PFS in ovarian cancer regardless of the treatment setting or type of drug used (VEGF inhibitors, VEGF-R inhibitors or angiopoietin inhibitors), thus being an option in the treatment of ovarian cancer. Secondly, due to the fact that OS improvements are only seen in high-risk chemo-naive cancers and in recurrent disease, we believe that angiogenesis inhibitors need to be administrated with prudence outside of these setting and after careful consideration of each case prognostic and response factors. Thirdly, we consider that further studies are required in order to better understand which categories benefit the most from angiogenesis inhibitors and which are only subjected to the risk of adverse effects and diminished QoL. The need to identify and validate the biomarkers for accurately predicting the response to therapy is a good future direction for study. Fourthly, for the platinum-sensitive recurrent disease, the role of concurrent therapy with angiogenetic and PARP inhibitors needs to be explored further. Lastly, we consider that the usage of a surrogate endpoint for clinical outcomes (PFS) in the process of drug approval may need to be reconsidered, as PFS proved not to be a viable substitute for OS.

## Figures and Tables

**Figure 1 diagnostics-13-01040-f001:**
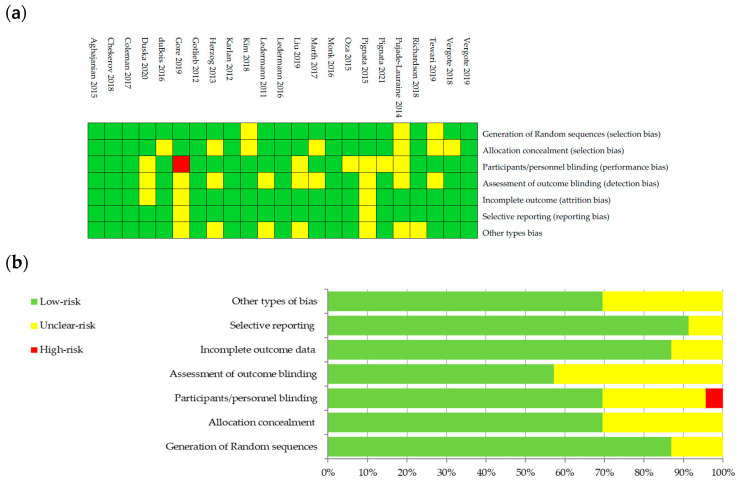
Graphical representation of bias risk: (**a**) authors’ judgements about each risk item color coded as low risk (green), unclear risk (yellow) or high risk (red) for studies included; and (**b**) percentage summary of bias risk across all studies, using same color coding of risk categories.

**Figure 2 diagnostics-13-01040-f002:**
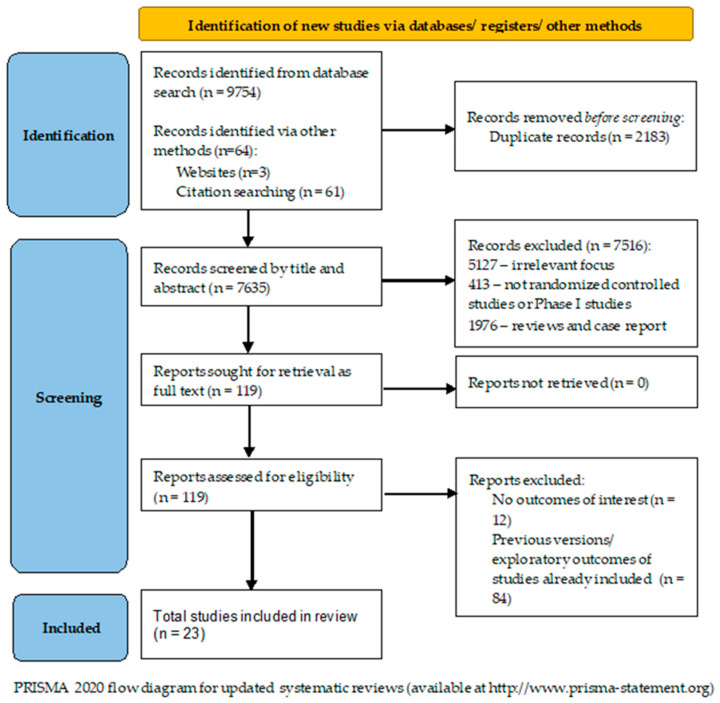
The literature search flow diagram.

**Figure 3 diagnostics-13-01040-f003:**
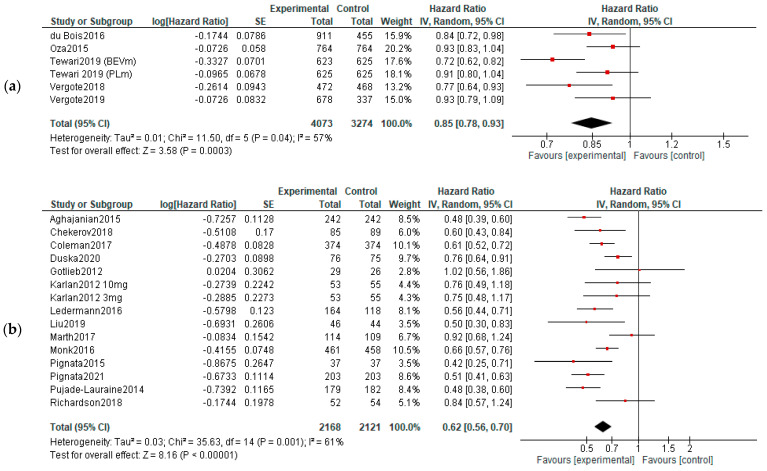
Settings subgroup analysis—time to progression: (**a**) first line, (**b**) recurrent, (**c**) P-S R, (**d**) P-R R and (**e**) maintenance. The black diamonds are visual representations of pooled effect and its confidence intervals and are non-significant if they overlap 1. Red squares represent each study effect size and study weight (size of square). The lines represent each study confidence interval. PLm—placebo maintenance; BEVm—bevacizumab maintenance [26,27,28,29,30,33,34,35,36,37,38,39,40,41,42,43,44,45,46,47,48,50,52].

**Figure 4 diagnostics-13-01040-f004:**
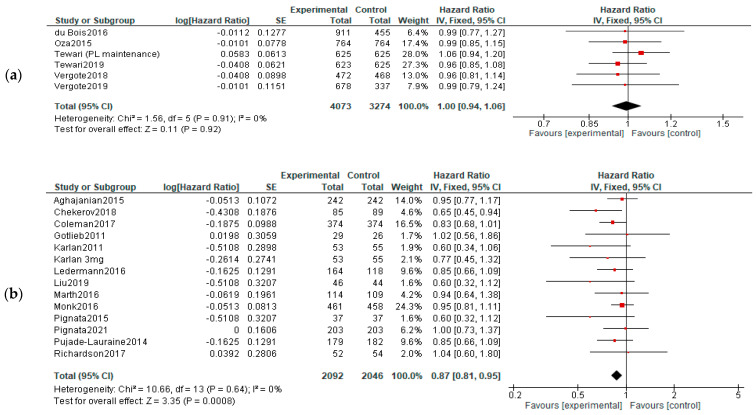
Settings subgroup analysis of overall survival: (**a**) first line, (**b**) recurrent, (**c**) P-S R, (**d**) P-R R and (**e**) maintenance. The black diamonds are visual representations of the pooled effect and its confidence intervals and are non-significant if they overlap 1. Red squares represent each study effect size and study weight size of square). The lines represent each study confidence interval. PLm—placebo maintenance; BEVm—bevacizumab maintenance [26,27,28,29,30,34,35,36,37,38,39,40,41,42,43,44,45,46,47,48,50,52].

**Figure 5 diagnostics-13-01040-f005:**
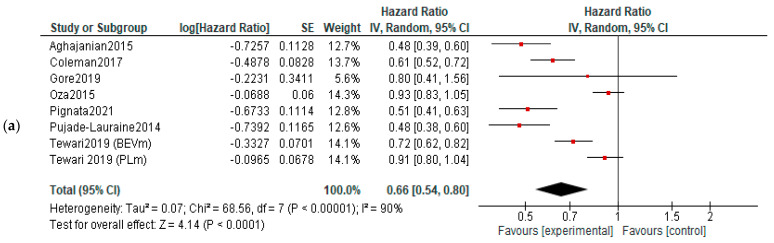
Action-mechanisms subgroup analysis of progression: (**a**) VEGF inhibitors, (**b**) VEGF-R inhibitors and (**c**) angiopoietin inhibitors. The black diamonds are visual representations of pooled effect and its confidence intervals and are non-significant if they overlap 1. Red squares represent each study effect size and study weight (size of square). The lines represent each study confidence interval. PLm—placebo maintenance; BEVm—bevacizumab maintenance [26,27,28,29,30,32,34,35,36,37,38,39,40,41,42,43,44,45,46,47,48,50,52].

**Figure 6 diagnostics-13-01040-f006:**
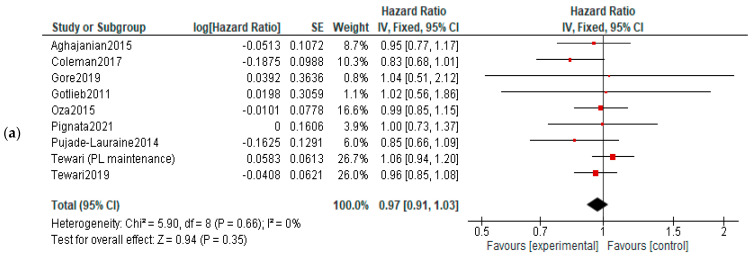
Action-mechanisms subgroup analysis of overall survival: (**a**) VEGF inhibitors, (**b**) VEGF-R inhibitors and (**c**) angiopoietin inhibitors. The black diamonds are visual representations of pooled effect and its confidence intervals and are non-significant if they overlap 1. Red squares represent each study effect size and study weight (size of square). The lines represent each study confidence interval. PLm—placebo maintenance; BEVm—bevacizumab maintenance [26,27,28,29,30,31,32,33,34,35,36,37,38,39,40,41,42,43,44,45,46,47,48,50,52].

**Figure 7 diagnostics-13-01040-f007:**
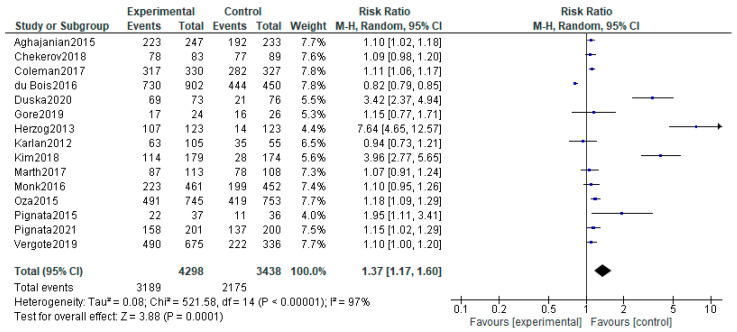
Grade ≥ 3 adverse effects associated with antiangiogenetic therapy. The black diamonds are visual representations of pooled effect and its confidence intervals and are non-significant if they overlap 1. Blue squares represent each study effect size and study weight (size of square). The lines represent each study confidence interval [26,27,28,29,30,32,34,35,36,41,42,43,44,45,52].

**Table 1 diagnostics-13-01040-t001:** PICOS criteria for inclusion of trials.

Parameter	Inclusion Criteria
Participants	Adults with confirmed ovarian cancer
Intervention	Angiogenesis-inhibitor therapy
Comparison	Drug regimens without angiogenesis inhibitors
Outcomes	PFS (hazard ratio, HR; confidence interval, 95% CI), OS (HR and 95% CI) and adverse effects (toxicity)
Study design	Randomized/controlled trials

PFS—progression-free survival; HR—hazard ratio; CI—confidence interval; OS—overall survival.

**Table 2 diagnostics-13-01040-t002:** General characteristics of RCTs included.

Study (Reference/Name/Phase)	Drug	Subjects	Sample Size (E/C)	Angiogenesis Inhibitors Group Treatment	Control Group Treatment	Outcomes in Meta-Analysis
Aghajanian2015 (OCEANS/ NCT00434642) Phase III [26]	Bevacizumab	P-S R EOC/fallopian/primary peritoneal carcinoma ECOG performance status PS 0–1	242/242	Cycles 1–6: gemcitabine (1000 mg/m^2^, days 1 and 8) + carboplatin (AUC 4, day 1) + bevacizumab (15 mg/kg on day 1, 6–10 cycles of 21 days) Cycles 10+: bevacizumab (15 mg/kg)	Cycles 1–6: gemcitabine (1000 mg/m^2^, days 1 and 8) and carboplatin (AUC4, day 1) + Placebo (15 mg/kg days 1, 6–10 cycles of 21 days) Cycles 10+: placebo (15 mg/kg)	PFS; OS; toxicity
Chekerov2018 (TRIAS/ NCT01047891) Phase II [27]	Sorafenib	P-R R EOC/fallopian/peritoneal carcinomas progressing during platinum therapy (platinum refractory) or ≤6 months after completing primary/secondary/ tertiary platinum-based therapy ECOG PS 0–2	85/89	Cycles 1–6: topotecan (1–25 mg/m^2^ on days 1–5) + sorafenib (400 mg oral bi-daily on days 6–15, every 21 days) Cycles 6+: maintenance sorafenib for up to 1 year daily	Cycles 1–6: topotecan (1–25 mg/m^2^ on days 1–5) + placebo (bi-daily on days 6–15, every 21 days) Cycles 6+: maintenance Placebo for up to 1 year daily	PFS; OS; toxicity
Coleman2017 (GOG-0213/ NCT00565851) Phase III [28]	Bevacizumab	P-S R EOC/fallopian/primary peritoneal cancer GOG PS 0–2	337/337	Cycles 1–6: paclitaxel (175 mg/m^2^)–carboplatin (AUC 5)) 3-weekly+ bevacizumab (15 mg/kg, 3-weekly) Cycles 6+: bevacizumab (15 mg/kg, 3-weekly)	Cycles 1–6: paclitaxel (175 mg/m^2^)–carboplatin (AUC 5) 3-weekly	PFS; OS; toxicity
Duska2020 (NCT01610206) Phase III [29]	Pazopanib	FIGO II-IV epithelial ovarian, fallopian tube or primary peritoneal carcinoma with less than 3 lines of prior chemotherapy ECOG PS 0–1	73/75	Gemcitabine (1000 mg/m^2^ every week on days 1 and 8, every 21 days with pazopanib 800 mg daily	Gemcitabine (1000 mg/m^2^ every week on days 1 and 8, every 21 days	PFS; toxicity
duBois2016 (AGO-OVAR12/NCT 01015118) Phase III [30,31]	Nintedanib	Chemo-naive, FIGO IIB-IV EOC/fallopian/primary peritoneal cancer ECOG performance status 0–2	911/455	Cycles 1–6: paclitaxel (175 mg/m^2^) + carboplatin (AUC5 or 6) + nintedanib (200 mg bi-daily, days 2–21), every 3 weeks followed by nintedanib maintenance	Cycles 1–6: paclitaxel (175 mg/m^2^) + carboplatin (AUC5 or 6) + placebo (200 mg, bi-daily, days 2–21, every 3 weeks), followed by placebo maintenance	PFS
Gore2019 (GOG-0241/ NCT01081262) Phase III [32]	Bevacizumab	Primary mEOC FIGO II–IV/ recurrence after stage I cancer	24/26	Paclitaxel (175 mg/m^2^) + carboplatin (AUC5/6) + bevacizumab (15 mg/kg, 3-weekly maintenance, 12 cycles). oxaliplatin (130 mg/m^2^) + capecitabine (850 mg/m^2^, bi-daily, days 1–14) + bevacizumab (15 mg/kg, 3-weekly maintenance, 12 cycles)	Paclitaxel (175 mg/m^2^) + carboplatin (AUC 5/6); Oxaliplatin (130 mg/m^2^) + capecitabine (850 mg/m^2^, bi-daily, days 1–14)	PFS; OS; toxicity
Gotlieb2012 (NCT00327444) Phase II [33]	Aflibercept	Platinum and Topotecan resistant and/or PLD-resistant cancer; Advanced EOC patients with recurrent malignant ascites ECOG Performance status 0–2	26/29	Aflibercept (4 mg/kg, every 2 weeks)	Placebo (4 mg/kg, every 2 weeks)	OS; toxicity
Herzog2013 (NCT00791778) Phase II [34]	Sorafenib	FIGO III–IV EOC/primary peritoneal cancers who responded after standard first-line platinum/taxanes containing chemotherapy ECOG Performance status 0–1	123/123	Sorafenib (400 mg bi-daily, every 12 h)	Placebo (400 mg bi-daily, every 12 h)	PFS; OS; toxicity
Karlan2012 (10 mg/kg) (NCT00479817) Phase II [35]	Trebananib (AMG 386)	FIGO II-IV recurrent EOC/ fallopian/primary peritoneal cancer ECOG Performance status 0–1	53/55	Paclitaxel (80 mg/m^2^ once a week, 3 weeks on/1 week off) + AMG 386 (10 mg/kg, once a week)	Paclitaxel (80 mg/m^2^ once a week, 3 weeks on/1 week off) + placebo (10 mg/kg, once a week)	PFS; OS; toxicity
Karlan2012 (3 mg/kg) (NCT00479817) Phase II [35]	Trebananib (AMG 386)	FIGO II-IV recurrent EOC/ fallopian/primary peritoneal cancer ECOG Performance status 0–1	53/55	Paclitaxel (80 mg/m^2^ once a week, 3 weeks on/1 week off) + AMG 386 (3 mg/kg, once a week)	Paclitaxel (80 mg/m^2^ once a week, 3 weeks on/1 week off) + placebo (3 mg/kg, once a week)	PFS; OS; toxicity
Kim2018 (East Asian Study/ NCT00866697) Phase III [36]	Pazopanib	Advanced EOC/fallopian/ primary peritoneal carcinoma	73/72	Pazopanib 800 mg daily for up to24 months	Placebo 800 mg daily or up to 24 months	PFS; OS
Ledermann2011 (NCT00710762) Phase II [37]	Nintedanib (BIBF 1120)	Advanced recurrent serous ovarian/fallopian/primary peritoneal cancer which responded to second-/third-/fourth-line chemotherapy ECOG Performance status 0–1	43/40	Cycles 1–9: BIBF 1120 (250 mg, bi-daily, 28-day cycles)	Cycles 1–9: placebo (250 mg, bi-daily, 28-day cycles)	PFS; OS; toxicity
Ledermann2016 (ICON6/ NCT00532194) Phase III [38,39]	Cediranib	P-S R EOC/fallopian/primary peritoneal cancer after first-line platinum-based chemotherapy ECOG Performance status 0–1	164/118	Platinum-based chemotherapy + cediranib (20 mg, daily) and then maintenance cediranib (20 mg, daily) alone	Platinum-based chemotherapy + placebo (20 mg, daily) then maintenance placebo (20 mg, daily)	PFS; OS; toxicity
Liu2019 (NCT01116648) Phase II [40]	Cediranib	P-S R high-grade serous/ endometrioid/deleterious germline BRCA1/2 mutation ovarian cancer	46/44	Olaparib (200 mg, bi-daily) + cediranib (30 mg daily)	Olaparib (400 mg, bi-daily)	PFS; OS; toxicity
Marth2017 (TRINOVA-2/ NCT01281254) Phase III [41]	Trebananib (AMG 386)	P-R R EOC/fallopian/primary peritoneal cancer ECOG Performance status 0–2	114/109	Pegylated liposomal doxorubicin (50 mg/m^2^, every 4 weeks) + trebananib (15 mg/kg, every week)	Pegylated liposomal doxorubicin (50 mg/m^2^, every 4 weeks) + placebo (15 mg/kg, every week)	PFS; OS; toxicity
Monk2016 (TRINOVA-1/ NCT01204749) Phase III [42]	Trebananib (AMG 386)	Recurrent partially platinum-sensitive or—EOC/fallopian/ primary peritoneal cancer GOG Performance status 0–1	458/461	Paclitaxel (80 mg/m^2^ once a week, 3 weeks on/1 week off) + trebananib (15 mg/ kg, every week)	Paclitaxel (80 mg/m^2^ once a week, 3 weeks on/1 week off) + placebo (15 mg/kg, every week)	PFS; OS; toxicity
Oza2015 (ICON7/ NCT00483782) Phase III [43]	Bevacizumab	FIGO I–IIA newly diagnosed high risk ovarian cancer/FIGO IIB–IV EOC/fallopian/primary peritoneal cancer ECOG Performance status 0–2	764/764	Cycles 1–6: paclitaxel (175 mg/m^2^) + carboplatin AUC 5 or 6) every 3 weeks + bevacizumab (7.5 mg/kg, every 3 weeks) Cycles 7–18: Bev (7.5 mg/kg, every 3 weeks)	Cycles 1–6: paclitaxel (175 mg/m^2^) + carboplatin AUC 5 or 6) every 3 weeks	PFS; OS; toxicity
Pignata2021 (MITO16b/ NCT01802749) Phase II [44]	Bevacizumab	FIGO IIIB-IV recurrent ovarian cancer relapsing ≥ 6 months after last dose of platinum, in patients with bevacizumab during first line treatment ECOG Performance status 0–2	203/203	Cycles 1–6: platinum-based-doublets paclitaxel–carboplatin/carboplatin–gemcitabine/carboplatin–pegylated liposomal doxorubicin + bevacizumab maintenance	Cycles 1–6: platinum-based doublets paclitaxel–carboplatin/carboplatin–gemcitabine/carboplatin–pegylated liposomal doxorubicin	PFS; OS; toxicity
Pignata2015 (MITO11/ NCT01644825) Phase II [45]	Pazopanib	Platinum-resistant/refractory EOC ECOG Performance status 0–1	37/37	Paclitaxel (80 mg/m^2^ on days 1, 8, and 15, every 28 days) + pazopanib 800 mg daily	Paclitaxel (80 mg/m^2^ on days 1, 8 and 15 every, 28 days)	PFS; OS; toxicity
Pujade-Lauraine 2014 (AURELIA/ NCT00976911) Phase III [46]	Bevacizumab	P-R R EOC/fallopian/primary peritoneal cancer ECOG Performance status 0–2	182/179	Cycle 1 to progression: pegylated liposomal doxorubicin (40 mg/m^2^, day 1 every 4 weeks) or paclitaxel (80 mg/m^2^, days 1, 8, 15, and 22, every 4 weeks); or topotecan (4 mg/m^2^, days 1, 8, and 15, every 4 weeks; or 1.25 mg/m^2^, days 1–5, every 3 weeks); + bevacizumab (10 mg/kg, every 2 weeks or 15 mg/kg, every 3 weeks)	Cycle 1 to progression: pegylated liposomal doxorubicin (40 mg/m^2^, day 1, every 4 weeks); paclitaxel (80 mg/m^2^ on days 1, 8, 15 and 22, every 4 weeks); or topotecan (4 mg/m^2^, days 1, 8, and 15, every 4 weeks; or 1.25 mg/m^2^, days 1–5, every 3 weeks);	PFS; OS; toxicity
Richardson2018 (NCT01468909) Phase II [47]	Pazopanib	Persistent or recurrent EOC/ fallopian/primary peritoneal cancer GOG Performance status 0–1	52/54	Paclitaxel (80 mg/m^2^ on days 1, 8, and 15 every 28 days) + pazopanib 800 mg daily	Paclitaxel (80 mg/m^2^ on days 1, 8 and 15 every 28 days) + placebo 800 mg daily	PFS; OS; toxicity
Tewari2019 (PLm) (GOG-0218/ NCT00262847) Phase III [48,49]	Bevacizumab	Newly diagnosed EOC/ fallopian/primary peritoneal cancer	625/625	Cycles 1–6: paclitaxel (175 mg/m^2^) + carboplatin (AUC 6) + bevacizumab (15 mg/kg; cycle 2 +) every 21 days Cycles 7–22: placebo maintenance every 21 days	Cycles 1–6: paclitaxel (175 mg/ m^2^) + carboplatin (AUC6) + PL (cycle 2 +) every 21 days Cycles 7–22: placebo maintenance every 21 days	PFS; OS; toxicity
Tewari2019 (BEVm) (GOG-0218/ NCT00262847) Phase III [48,49]	Bevacizumab	Newly diagnosed EOC/ fallopian/primary peritoneal cancer	623/625	Cycles 1–6: paclitaxel (175 mg/m^2^) + carboplatin (AUC 6) + bevacizumab (15 mg/kg; cycle 2 +) every 21 days Cycles 7–22: bevacizumab maintenance (15 mg/kg) every 21 days	Cycles 1–6: paclitaxel (175 mg/m^2^) + carboplatin (AUC 6) Placebo (cycle 2+) every 21 days Cycles 7–22: placebo every 21 days	PFS; OS; toxicity
Vergote2018 (AGO-OVAR16/ NCT00866697) Phase III [50,51]	Pazopanib	Newly diagnosed advanced ovarian cancer	472/468	Pazopanib 800 mg daily for up to 24 months	Placebo 800 mg daily for up to 24 months	PFS; OS; toxicity
Vergote2019 (TRINOVA-3/NCT01493505) Phase III [52]	Trebananib (AMG 386)	FIGO III–IV EOC/fallopian/ primary peritoneal cancer ECOG Performance status 0–1	678/337	Cycles 1–6: Pac (175 mg/m^2^)- carboplatin ((AUC 5/6) every 3 weeks) + trebananib (15 mg/kg) Cycles 6+: trebananib for up to 18 more months	Cycles 1–6: paclitaxel (175 mg/m^2^)–carboplatin ((AUC5/6) every 3 weeks) + placebo (15 mg/kg) Cycles 6+: placebo for up to 18 more months	PFS; OS; toxicity

P-S R—platinum-sensitive recurrent cancer; ECOG—Eastern Cooperative Oncology Group; PS—performance status; EOC—epithelial ovarian cancer; AUC—area under curve; PFS—progression free survival; OS—overall survival; FIGO—International Federation of Gynecology and Obstetrics; GOG—gynecological oncology group; P-R R—platinum-resistant recurrent cancer; mEOC—mucinous epithelial ovarian cancer; PLm—placebo maintenance; BEVm—bevacizumab maintenance.

**Table 3 diagnostics-13-01040-t003:** Summary of outcomes of included trials.

Study	Line	Size	Arms	PFS		OS	
				Median (in Months)	Hazard Ratio (95% CI)	Median (in Months)	Hazard Ratio (95% CI)
Aghajanian 2015 [26]	P-S R	484	GC + Pl + bevacizumab(m) GC + PL	12.4 8.4	0.484 (0.388–0.605)	33.6 32.9	0.95 (0.77–1.77)
Chekerov 2018 [27]	P-R R	174	Topotecan + sorafenib + sorafenib(m) PL + PL(m)	6.7 4.4	0.60 (0.43–0.83)	17.1 10.1	0.65 (0.45–0.93)
Coleman 2017 [28]	P-S R	674	GC + PL + bevacizumab(m) TC	13.8 10.4	0.628 (0.534–0.739)	42.2 37.3	0.829 (0.683–1.005)
Du Bois 2016 [30]	F	1366	TC +nintedanib + nintedanib(m) TC + PL + PL(m)	17.2 16.6	0.84 (0.72–0.98)	34 32.8	0.99 (0.77–1.27)
Duska 2020 [29]	R	148	Pazopanib PL	5.3 2.9	1.50 (0.76–2.94)	NA	NA
Gore 2019 [32]	F or R	50	TC/Oxal–Cape + Bev TC/Oxal–Cape	18.1 8.8	0.80 (0.41–1.57)	27.7 32.7	1.04 (0.51–2.10)
Gotlieb 2012 [33]	R	55	Aflibercept PL	6.3 w 7.3 w	NA	12.9 w 16.0 w	1.02 (0.56–1.86)
Herzog 2013 [34]	M	246	Sorafenib PL	12.7 15.7	1.09 (0.72–1.63)	NA	1.48 (0.69–3.23)
Karlan 2012 [35] (10 mg/kg)	R	108	Pac + trebananib Pac + PL	7.2 4.6	0.76 (0.49–1.18)	22.5 20.9	0.60 (0.34–1.06)
Karlan 2012 [35] (3 mg/kg)	R	108	Pac + trebananib Pac + PL	5.7 4.6	0.75 (0.48–1.17)	20.4 20.9	0.77 (0.45–1.31)
Kim 2018 [36]	M	145	Pazopanib PL	18.1 18.1	0.984 (0.596–1.626)	NA	0.811 (0.376–1.751)
Ledermann 2016 [38,39]	P-S R	282	TC or GC or Carbo + cediranib + cediranib(m) TC or GC or Carbo + PL +PL(m)	11.0 8.7	0.56 (0.44–0.72)	27.3 19.9	0.85 (0.66–1.10)
Ledermann 2011 [37]	M	83	Nintedanib PL	NA	0.65 (0.41–1.02)	NA	0.84 (0.51–1.39)
Liu 2019 [40]	P-S R	90	Olaparib + cediranib Olaparib	16.5 8.2	0.5 (0.3–0.83)	44.2 33.3	0.64 (0.36–1.11)
Marth 2017 [41]	P-S R	223	PLD + trebananib PLD + PL	7.6 7.2	0.92 (0.68–1.24)	19.4 17.0	0.94 (0.64–1.39)
Monk 2016 [42]	R	919	Pac + trebananib Pac + PL	7.2 5.4	0.66 (0.57–0.77)	19.3 18.3	0.95 (0.81–1.11)
Oza 2015 [43]	F	1528	TC + Bev + Bev(m) TC	19.9 17.5	0.93 (0.83–1.05)	58.0 58.6	0.99 (0.85–1.14)
Pignata 2015 [45]	P-R R	73	Pac + pazopanib Pac + PL	6.35 3.49	0.42 (0.25–0.69)	19.1 13.7	0.60 (0.32–1.13)
Pignata 2021 [44]	P-S R	406	TC/GC/Carbo–PLD + Bev TC/GC/Carbo–PLD	11.8 8.8	0.51 (0.41–0.64)	26.7 27.1	1.00 (0.73–1.39)
Pujade-Lauraine 2014 [46]	P-R R	361	PLD/Pac/TOP + Bev PLD/Pac/TOP	6.7 3.4	0.48 (0.38–0.60)	16.6 13.3	0.85 (0.66–1.08)
Richardson 2018 [47]	R	106	Paclitaxel + pazopanib Paclitaxel + PL	7.5 6.2	0.84 (0.57–1.22)	20.7 23.3	1.04 (0.60–1.79)
Tewari 2019 [48] (PLm)	F	1250	TC + bevacizumab + PL(m) TC + PL	11.2 10.3	0.908 (0.795–1.040)	40.8 41.1	1.06 (0.94–1.20)
Tewari 2019 [48] (BEV-m)	F	1248	TC + bevacizumab + bevacizumab(m) TC + PL	14.1 10.3	0.717 (0.625–0.824)	43.4 41.1	0.96 (0.85–1.09)
Vergote 2018 [50]	F	940	Pazopanib PL	17.9 12.3	0.77 (0.64–0.91)	59.1 64.0	0.96 (0.805–1.145)
Vergote 2019 [52]	F	1015	TC + trebananib + trebananib(m) TC + placebo + placebo (m)	15.9 15.0	0.93 (0.79–1.09)	46.6 43.6	0.99 (0.79–1.25)

F—first line; R—recurrent; P-R R—platinum-resistant recurrent; P-S R—platinum-sensitive recurrent; TC—paclitaxel + carboplatin, Oxal—oxaliplatin; Cape—capecitabine; Bev—bevacizumab; m—maintenance therapy; PL—placebo; GC—gemcitabine + carboplatin; Carbo—carboplatin; PLD—pegylated liposomal doxorubicin; Pac—paclitaxel; TOP—topotecan; w—weeks; NA—not available; M—pure maintenance.

## Data Availability

The data is available online in the references we included. There is no outside storage of data since we did not use any data of our own and made a statistical analysis of data published by other authors.

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
