# Peer review of "Analysis of Efficacy-To-Safety Ratio of Angiogenesis-Inhibitors Based Therapies in Ovarian Cancer: A Systematic Review and Meta-Analysis"

_diagnostics, 2023, doi:10.3390/diagnostics13061040_

Round 1
Reviewer 1 Report
This manuscript describes a systematic review and meta-analysis of 23 randomized controlled trials (12081 patients). The overall survival – OS, progression free survival– PFS and toxicity (grade ≥3 adverse effects, type and number of all adverse effects) were evaluated. The analyzed data showed improvement of pooled-PFS in first-line or recurrent cancer and regardless of type of anti-angiogenesis drug used. Improved OS was also observed but only in recurrent platinum-sensitive or platinum-resistant cancers. The analysis was carried out accurately and underline the given conclusions. I recommend acceptance after minor revision:
1. Some grammatical and typing errors are seen throughout the manuscript, therefore, these errors should be corrected before acceptance. The suggested corrections are as comments in the attached pdf manuscript.

Author Response
Regarding reviewer 1, I can only thank him/her for the way he/she understood and appreciated our work and assure him/her that in the revised form of this manuscript we have taken into account every suggestion he/she made to us, trying to bring that recommended “polish” to the manuscript. The manuscript has been revised extensively and I am convinced that the improved version will receive the same appreciation from the reviewer, to whom I thank once again.
Reviewer 2 Report
Dear Author(s)
1. There are grammatical errors.
2. "The following combinations of search terms" is unclear.
3. The flowchart is not based on PRISMA.
4. Please add the full name of the abbreviation for the first time such as VEGF, VEGFR, VEGF-R, ...
5. Please describe outcomes such as PFS and OS, ...
6. If the study in figures returns to references. Please add the number of references to each figure. Also, add the publication year after the author's name for each study in each figure.
7. Please add a description for the black diamond, red square, ... for the forest plots below them.
8. You have a mistake in Figure 7. The entering data for the study of Duska is incorrect. Please re-edit it.
9. Please add the full name of each abbreviation below each table.
10. Please add meta-regression, TSA, radial plot, senstivity analyses, ...
11. THere is a meta-analysis related to the subject. Please update the results and what is the novelty of your work compared to the previous one? https://www.ncbi.nlm.nih.gov/pmc/articles/PMC9396859/
Author Response
Regarding reviewer 2, I must admit that he gave us work. But we thank him/her all the more, because the critical aspects he/she addressed helped us enormously to improve the manuscript.
So, addressing reviewer number 2 directly, we, the authors, would like to express how deeply honored we are that our work was submitted to such a rigorous and extremely patient reviewer. We are delighted that he/she have dedicated serious time to an in-depth analysis of our manuscript and we thank for the suggestions which will allowed us to improve our work and produce relevant material. After reviewing our material in accordance with these suggestions, we also found a few additional small mistakes that we were able to rectify in time, and we are grateful for this contribution. We have addressed each of your concerns as follows:
- Suggestion 1: Grammatical errors - The document was subjected to grammar and spelling checks and revised by another of our authors which has a Native Speaking Level English Language degree. Furthermore, if deemed necessary, we will use professional checking of our manuscripts before printing.
- Suggestion 2: "The following combinations of search terms" is unclear. We have revised the entire section “Search strategy” and provided additional information about the syntax and criteria used for database searches.
- Suggestion 3: “The flowchart is not based on PRISMA”. – Our study follows the PRISMA guidelines.
- We have updated our literature search flowchart according to PRISMA 2020 statement (available online at https://www.bmj.com/content/372/bmj.n71: Page M J, McKenzie J E, Bossuyt P M, Boutron I, Hoffmann T C, Mulrow C D et al. The PRISMA 2020 statement: an updated guideline for reporting systematic reviews BMJ 2021; 372 :n71 doi:10.1136/bmj.n71) and inserted it as a Figure 1.
- We also included a PRISMA checklist as supplementary material together with our article.
- Suggestion 4: We added full name of the abbreviations for the first-time use – VEGF, VEGF-R, PARP, PFS, OS and so on. We did not include full name for the standard statistical abbreviations like HR (hazard ratio) or CI (confidence interval) or RR (risk ratio).
- Suggestion 5: We included a brief description of reported outcomes for each study in Table 3. Summary of outcomes of included trials. In regard to our results the pooled PFS and OS are reported in text Subsection 3.3: Analysis of survival: overall and progression-free. In the same subsection we inserted results of PFS and OS subgroup analysis (both for treatment settings categories and types of drugs used). We added subtitles for each outcome. Adverse events results are described in text in Subsection 3.4. Adverse events. We reported results for 31 of the secondary effects known to occur during treatment with anti-angiogenesis inhibitors, analyzed as risk ratios. Forest-plots for each effect is shown on Supplementary Materials – Figures S1-S31. There is also a large section in Section 4: Discussions, interpreting our results in context of previous studies, pointing out possible explanations for results, clinical correlations, and future directions of research.
- Suggestion 6: We have added the publication year of each study after the author’s name in each figure. We added references in all tables and after each figure.
- Suggestion 7: We added a description for the black diamonds, red squares, blue squares and lines in the forest plots and provided a short description on how to read them below each figure.
- Suggestion 8: We have corrected the mistake in Figure 7 for the Duska study and reedited the figure. We also corrected the portion of text regarding Figure 7, to reflect the statistic change.
- Suggestion 9: We added the full name of each abbreviation below each table and figure.
- Suggestion 10: We added sensitivity analysis and Galbraith plots as supplementary material and referred it in text.
- Suggestion 11: Our search included literature until 1st May 2022, and we did not come across this study. We read the full article you indicated and found it very well put together and relevant to our study, even if it only studies a subset of ovarian cancers (recurrent disease). We included a reference to this study in the biography section of our manuscript and dedicated a paragraph highlighting the differences between that study and ours.
Round 2
Reviewer 2 Report
Dear
1. There are several grammatical errors yet.
2. What is the reasons of heterogeneity? Subgroup analysis. TSA. Meta-regression.
3. You add radial plots without complete derails about it in methods and results.
4. The analysis is unclear because of corrections with text.
5. Please add "p"in p-value as italic.
6. Was there limitations in searching.
7. Where is "2.3." in methods?
8. Where is statistical analysis?
9. Figure 1 is belongs to Results, not Methods.
Author Response
- The manuscript was subjected to multiple automated and manual grammar and spelling checks and it also has been revised by another of our authors which holds a Native Speaking Level English Language degree. As stated before, if deemed necessary, we will use professional checking by MDPI for our manuscripts before printing.
- A paragraph better explaining the reasons for heterogeneity was added to Subsection 4.3. Strengths and study limitations.
Extensive subgroup analysis by treatment settings, by platinum sensitivity of disease and by type of drug for each primary outcome of interest used (OS-overall survival and PFS- progression free survival) is provided in Subsection 3.3 Analysis of survival: overall and progression-free.
We would like to point out that none of the meta-analysis published until now about anti-angiogenetic drugs in ovarian cancer that we know of, including the one you suggested before, do not include trial sequential analysis (TSAs), and because of that we did not consider it necessary to include one in ours. In a rapid search of PubMed database from 1st of January 2014 and 28 February 2023 using as search terms ((meta-analysis OR metaanalysis OR meta analysis) and (TSA OR Trial Sequential Analysis)), we could only find 10 articles containing TSAs (most of them having very small number of participants or very few studies included) among more than 120.000 meta-analysis studies published in the same time period. In our case, the number of patients included in our study is very large (more than 12000 patients) and we have 22 trials included, thus limiting the potential type I (false-positive conclusions) or type II (false-negative conclusions) errors that can arise from insufficient number of participants, especially in the pooled results. Some errors may still occur in certain subgroups where the number of patients is limited due to lack of additional studies. The need for further studies on certain subgroups was explained in the text of our manuscript. TSA can be used to add strength to the meta-analytic conclusions by estimating if the effect is large enough to be unlikely affected by further studies. However, pre-registration of TSA protocols is paramount. We did not register our meta-analysis, nor did we register a TSA protocol. We acknowledge this as a limitation of our study and state so in Subsection 4.3. Strengths and study limitations.
We did not perform a meta-regression analysis because, as you well know, although apparently very fancy, this method has several serious theoretical and practical limitations that can impair the model’s ability to make valid inferences. First, sample-size may be insufficient for analysis. Second, some estimation methods based on asymptotical assumptions can easily be biased when the sample size is small. These limitations can result in inability to adjust for confounding, and even if the information on confounders is present and the number of studies included is moderate to large, the characteristics of the studies included tend to be corelated, thus creating a collinearity problem. Another limitation inherent to the method is that it is subject to ecological fallacy risk since meta-regression makes inference about individuals from trial-level information. Finally, since it is based on literature reviews, the results are susceptible to publication bias, quantitative measurements being especially prone to false positive findings. In conclusion, a meta-regression analysis is not always necessary (low heterogeneity between studies) and needs to be performed with extreme caution since it is a method prone to error, bias, poor implementation and misinterpretation of results.
- References to Galbraith plots were made in subsection 3.3 Analysis of survival: overall and progression-free - Heterogeneity and publication bias, describing their role in the analysis. We do not consider necessary additional explanation on how to read radial plots, since this is a paper addressed to academics that do possess a basic understanding of graphical representations of statistical analysis.
- We removed the highlighted changes we made to the manuscript to make it more readable.
- We made Italic all Ps in p-value.
- Search was limited to journals and articles published in English. We added explanation to 2.1. Search strategy.
- Subsection 2.3. Data extraction, although existing, was just misaligned. We did the proper adjustment.
- There is a large section on statistical results obtained, in 3.3 Analysis of survival: overall and progression-free and 3.4. Adverse events providing statistical analysis and results obtained. There is also Section 4: Discussions, interpreting our results in context of previous studies, pointing out possible explanations for results, clinical correlations, and future directions of research.
- As suggested we moved Figure 1 in Results.